# A New Paclitaxel Formulation Based on Secretome Isolated from Mesenchymal Stem Cells Shows a Significant Cytotoxic Effect on Osteosarcoma Cell Lines

**DOI:** 10.3390/pharmaceutics15092340

**Published:** 2023-09-19

**Authors:** Alessia Giovanna Santa Banche Niclot, Elena Marini, Ivana Ferrero, Francesco Barbero, Elena Rosso, Ivana Fenoglio, Alessandro Barge, Augusto Pessina, Valentina Coccè, Francesca Paino, Katia Mareschi, Franca Fagioli

**Affiliations:** 1Department of Public Health and Paediatrics, The University of Turin, Piazza Polonia 94, 10126 Torino, Italy; alessiagiovannasanta.bancheniclot@unito.it (A.G.S.B.N.); elena.marini@edu.unito.it (E.M.); franca.fagioli@unito.it (F.F.); 2Stem Cell Transplantation and Cellular Therapy Laboratory, Paediatric Onco-Haematology Division, Regina Margherita Children’s Hospital, City of Health and Science of Turin, 10126 Torino, Italy; ivana.ferrero@unito.it; 3Department of Chemistry, University of Turin, Via Pietro Giuria 7, 10125 Turin, Italy; francesco.barbero@unito.it (F.B.); ivana.fenoglio@unito.it (I.F.); 4Department of Drug Science and Technology, University of Turin, Via Pietro Giuria 9, 10125 Turin, Italy; elena.rosso@unito.it (E.R.); alessandro.barge@unito.it (A.B.); 5CRC StaMeTec, Department of Biomedical, Surgical and Dental Sciences, University of Milan, 20122 Milan, Italy; augusto.pessina@unimi.it (A.P.); valentina.cocce@unimi.it (V.C.); francesca.paino@unimi.it (F.P.)

**Keywords:** mesenchymal stem cells, secretome, drug delivery system, paclitaxel, osteosarcoma

## Abstract

Background: Osteosarcoma (OS) represents a rare cancer with an unfavorable prognosis that needs innovative treatment. The aim was to isolate a secretome from mesenchymal stem cells (MSCs) that are treated with paclitaxel (PTX)-containing microvesicles as a drug delivery system and analyze its cytotoxic effects on OS cell lines (SJSA, MG63, and HOS). Methods: Three batches of secretome (SECR-1, SECR-2, and SECR-3) were produced from three bone marrow (BM) MSCs samples treated for 24 h with 15 µg/mL of PTX or with a standard medium. The viability of the OS cell lines after 5 days of exposure to SECR-1-2-3 (pure and diluted to 1:2 and 1:4) was analyzed with an MTT assay. The same SECR batches were analyzed with high-performance liquid chromatography (HPLC) and with a nanoparticle tracking assay (NTA). Results: A statistically significant decrease in the viability of all OS cell lines was observed after treatment with SECR-PTX 1-2-3 in a dose–response manner. The NTA analyses showed the presence of nanoparticles (NPs) with a mean size comparable to that of extracellular vesicles (EVs). The HPLC analyses detected the presence of PTX in minimal doses in all SECR batches. Conclusions: This proof-of-concept study showed that the conditioned medium isolated from MSCs loaded with PTX had a strong cytotoxic effect on OS cell lines, due to the presence of EV and PTX.

## 1. Introduction

Sarcomas are rare mesenchymal tumors, occurring in the bones or soft tissues, which affect all ages but are more common in the pediatric age group. In fact, despite their rarity, they constitute significant mortality in around 13% of cancers [1]. Presently, osteosarcoma (OS) is a sub-type of sarcoma that is difficult to treat and has an unfavorable prognosis. Despite local tumor resection and intense chemotherapy treatment, OS is a very aggressive disease as 30–35% of patients have a local or systemic recurrence that evolves with an unfavorable prognosis: only 25% of patients survive more than five years [2]. The survival rate of patients is still lagging behind the overall survival rates of patients with other cancers in that age group, mainly due to intrinsic or acquired drug resistance [3] and the formation of lung metastases, which represent the disease’s primary cause of mortality [4]. Over the last decade, the therapeutic approach has been improved, but no innovative or efficient standardized therapy has been found to date. Standard therapy consists of a combination of surgery with radiotherapy and high-dose systemic chemotherapy, both before and after surgical resection of the tumor, but relapses and metastases often occur. In addition, systemic chemotherapy can sometimes induce multidrug resistance and side effects such as cardiotoxicity or nephrotoxicity [5]. Since standard chemotherapy is not conclusive and no effective therapeutic modalities have emerged, it is necessary to develop new therapeutic approaches and innovative drug delivery systems (DDS) in order to facilitate OS therapy. The target of chemotherapy treatment is cells with a rapid turnover, which is characteristic of both healthy cells and cancer cells. Therefore, it is essential to identify systems that are suitable for carrying anticancer drugs, which also show selectivity for cancer cells compared to healthy ones. To improve the performance of chemotherapy, an appropriate design for a drug-carrier system is needed. DDS are among the most innovative methods developed in the last few years of drug administration. DDS allow the controlled administration of drugs through nano-systems that direct the release of the molecules of interest to a specific cell type, thus circumscribing their biological effect. These characteristics work together to improve therapeutic efficacy and reduce pharmacological toxicity since the treatment does not act systemically, thus avoiding compromising healthy organs. A type of DDS that is usable in various pathologies, including neoplasms and, therefore, pediatric sarcomas such as OS, is a secretome derived from mesenchymal stem cells (MSCs). MSCs can be isolated from several tissues (e.g., adipose tissue, bone marrow, skin, umbilical cord blood, etc.). More importantly, MSCs are able to home in on inflammatory microenvironments and will also migrate to tumor masses after systemic injection; they offer a promising tool for DDS in the treatment of tumors [6]. Bone marrow (BM) MSCs, primed in vitro with paclitaxel (PTX), are able to incorporate significant amounts of the drug, which is subsequently released into the culture medium, as well as through exosome release [7,8]. Recent studies have shown that MSCs derived from different tissues (bone marrow, adipose tissue, and gingival papilla) are able to uptake and deliver drugs without any genetic manipulation -and drug-loaded MSCs can acquire significant anti-tumor activity due to their capacity to take up the drug and release it at the proper site [9,10]. This transport and delivery of anti-cancer drugs by MSCs offer several potential advantages with respect to the use of free drugs. For example, by increasing the level of protection given to the drug against degradation and enhancing the drug concentration introduced into the tumor mass, since MSCs are able to integrate into the tumor stroma due to the availability of loco-regional drug delivery, this will also increase the efficacy of treatment and contribute to reducing the systemic toxicity [11]. The above-reported studies suggest that drug delivery by MSCs can be a very useful approach in the treatment of many solid tumors. For this reason, the development of advanced therapy medicinal products (ATMPs) requires the optimization of cell culture conditions according to good manufacturing practice (GMP) [12] to obtain a high number of cells that are safe for application in cell therapy areas. On the basis of these recent reports, the Italian Drug Agency has approved the procedure of loading MSCs with PTX, using a large-scale bioreactor system for their storage until clinical use [13].

Recent studies have shown that the therapeutic biological effects of MSCs lie in the release of bioactive factors known as the secretome. This can be used as a possible alternative to cell therapy and also as a physiological carrier of anticancer agents, thanks to the extracellular vesicles (EVs) contained in the secretome itself, which reflects the genetic and proteomic contents of the secreting MSCs [14]. In addition, without genetic manipulation, the EVs can incorporate a significant degree of chemotherapy, which is subsequently released in sufficient quantities to influence the proliferation of cancer cells in both in vitro and in vivo molecules and, thanks to their physiological tropism, can reach the injured sites or tumor stroma. MSCs thus incorporated in the secretome could replace MSC therapy, offering safety, functionality, and storage, while limiting the tumorigenic risk associated with cell therapy. 

We aimed to develop a proof-of-concept (POC) study to verify the use of secretome-containing microvesicles loaded with drugs, which could be considered a new therapeutic approach to treating OS patients. 

For this purpose, we verified if the conditioned medium, harvested from MSCs loaded with PTX, contained a secretome with an antitumor effect on three OS cell lines (SJSA, MG63, and HOS); we demonstrated that it had a strong cytotoxic effect and showed the presence of microvesicles and PTX.

## 2. Materials and Methods

### 2.1. Bone Marrow Mesenchymal Stem Cells Secretome Production

Human bone marrow (BM) samples were collected after receiving a signature giving written consent, in accordance with the “City of Health and Science Hospital of Turin, Pediatric Onco Hematology, Regina Margherita Children’s Hospital” Ethics Committee, as required by the Declaration of Helsinki. Paclitaxel (Kiabi 6 mg/mL) is a Taxol derivative, and its mechanism of action is via interaction with microtubules, altering their polymerization and, thus, inhibiting cell mitosis. The drugs were used according to their drug dossier. BM cells were harvested from unfiltered BM collection waste bags, which are usually discarded after the BM infusion. The bag was washed with phosphate-buffered saline and the cells were collected and washed at 200 g for 5 min. The cells were cultured with Alpha Minimum Essential Medium Eagle (α-MEM, Sigma Aldrich, St. Louis, MO, USA), supplemented with 1% L-glutamine, 1% penicillin/streptomycin, and 10% of human platelet lysate (HPL) [15], and maintained at 37 °C in a 5% CO_2_ atmosphere. After 7–10 days, the non-adherent cells were removed and the adherent MSC cells were re-fed every 3–4 days. When the cells reached about 80% of confluence, they were detached with trypsin/EDTA (Sigma Aldrich), counted, re-seeded, and expanded. The cells at each passage were frozen in α-MEM with 5% albumin and 10% dimethyl sulfoxide (DMSO, Alchimia SRL, Ponte San Nicolò, Italy). The cells were cultured up to the third passage, then the MSC surface markers were analyzed by flow cytometry (Navios software version 1.2, Beckman Coulter, Krefeld, Germany), using the following antibodies: CD90 FITC, CD73 PE, CD105 PC7, CD146 APC, CD45-34–14 FITC, CD19 APC, and HLA-DR PE. At the same time, the cells were double-divided for the 2 conditions: (1) SECR-CTRL in standard medium and (2) SECR-PTX, where the cells were treated with PTX at a concentration of 15 µg/mL for 24 h, at 37 °C in a 5% CO_2_ atmosphere. After 24 h of PTX uptake, the medium was removed in both conditions and the MSCs were washed three times and cultured in serum-free medium for 48 h to induce EV release. At the end of serum starvation, the MSCs were counted, and their viability was evaluated by staining with Trypan Blue. The conditioned media was collected and then centrifuged at 3500× *g* for 15 min to eliminate cell debris and apoptotic bodies. Three batches of secretome (SECR-CTRL and SECR-PTX for each batch) were produced from three BM samples. The secretome thus obtained was used for a cytotoxicity assay on the OS tumor cell lines, nanoparticle tracking analysis (NTA), and high-performance liquid chromatography (HPLC).

### 2.2. Determination of PTX Sensitivity in MSCs and OS Cell Lines

First, we confirmed the literature data reporting the resistance of MSCs to high doses of paclitaxel. Three batches of BM-MSCs were expanded at the third passage; when they reached semi-confluence, they were treated with PTX at 15 µg/mL for 24 h. We then calculated the viability using Trypan Blue. Subsequently, for SJSA, MG63, and HOS, we calculated the PTX inhibitory concentration (IC50) using an MTT assay. The cells were cultivated and treated with scalar doses of PTX (from 320 µg/mL to 0 µg/mL) for 24 h. At the end of the time period, the cells were treated with 12 nM of MTT stock solution (as reported for the Vybrant MTT Cell Proliferation Assay Kit V-13154, Molecular Probes Europe BV, ThermoFisher, Waltham, MA, USA) and incubated at 37 °C for 4 h in the dark. After labeling with MTT, we added 50µL of DMSO and incubated it at 37 °C for 10 min. The absorbance was read at 540 nm, using a GloMax Discover Microplate Reader (Promega, Madison, WI, USA). The IC50 (Table 1) was calculated with the GraphPad statistical software, using nonlinear regression through standard curves.

### 2.3. Secretome Cytotoxicity Activities

The SECR-PTX cytotoxicity was tested using an MTT assay. The three OS cell lines (SJSA; MG63; HOS-2) were tested with the three secretome lots (SECR-1, SECR-2, SECR-3) and the test was performed in triplicate. The cells were plated at 15,000 cells/well in a 96-multiwell flask containing DMEM supplemented with 1% L-glutamine, 1% penicillin/streptomycin, and 10% of fetal bovine serum (FBS, Gibco, Waltham, MA, USA) for 24 h. After 24 h, the cells were treated with SECR-CTRL, SECR-PTX, and PTX, diluted in the standard medium with dilutions of 1:2 and 1:4, to study the dose-dependent response. The OS cells were treated with: (I) SECR-CTRL-1; SECR-PTX-1 (II) SECR-CTRL-2; SECR-PTX-2 (III) SECR-CTRL-3; SECR-PTX-3 (IV) PTX 15 µg/mL (V) Standard Medium, and were then incubated for 5 days at 37 °C in a 5% CO_2_ atmosphere. Finally, 12 nM of MTT stock solution was added to each well of the 96-multiwell flask and the flask was incubated for 4 h at 37 °C in the dark, as previously described. The absorbance was read at 540 nm, using a GloMax Discover Microplate Reader. The viability was calculated as a percentage by dividing the absorbance of the sample by the absorbance of the live control cells (cells treated with standard medium) × 100. The data were plotted using the GraphPad software and a statistical analysis was performed using a two-way ANOVA test.

### 2.4. Secretome Characterization by Nanoparticle Tracking Analysis

The analysis was performed using a ZetaView^®^ PMX-120 (Particle Metrix GmbH, Inning am Ammersee, Germany) nanoparticle tracking analysis (NTA) system, equipped with a light source wavelength of 488 nm. Before the measurements, the samples were diluted ten times in PBS to avoid changes in the osmolarity of the dispersion and to have an EV concentration suitable for the NTA analysis. After optimization of the instrumental parameters, the sensitivity and the shutter, set in scatter mode, were set at 70 and 100, respectively, in 3 × 33 videos of 1 s, recording a total number of particles of between 2000 and 3600 per analysis. In the fluorescent mode, the sensitivity and the shutter were set at 80 and 100, respectively. To stain the EVs, 1 uL of CellMask^TM^ Green (CMG) was added to the secretome and left in the dark at RT for 1 h, then the samples were diluted ten times in PBS. As a control, cell culture media with added CMG was used; no particles were detected under the same instrumental conditions.

### 2.5. PTX Determination in Secretome Preparation via HPLC-MS/MS Analysis

HPLC-MS/MS analyses were carried out on a Waters UPLC Acquity-TQD system equipped with an ACQUITY UPLC^®^ BEH C8 1.7 μm 2.1 × 50 column, using a binary mixture consisting of trifluoroacetic acid at 0.1% in water (eluent A) and trifluoroacetic acid at 0.1% in acetonitrile (eluent B).

The analysis was conducted in gradient according to the profile below, using a flow rate of 0.400 mL/min, an injection volume of 5 µL, and a column temperature of 40 °C. The autosampler temperature was kept at 20 °C.Gradient Profile (B%, min) was: 40, 0; 50, 3.47; 100, 4.34; 100, 6.00

The detector was set in MRM mode, following the m/z 854.50 → 286.19 transition with a cone potential of 25 V and a collision energy of 18 eV.

The system was calibrated with a series of PTX standards, with concentrations of between 50 ng/mL and 5 µg/mL, resulting in a straight line where r^2^ = 0.999614. The point at the lowest concentration (50 ng/mL) coincides with the LOQ and LOD values.

Sample preparation: The sample containing the secretome was frozen, then thawed and vortexed, and 10 mL of the sample was taken. To this sample, 10 mL of acetonitrile was added and the mixture was vortexed for 1 min. The mixture was then placed in an ultrasonic bath for 10 min and centrifuged at 8700 rpm for 5 min.

After centrifugation, the supernatant was removed and 10 mL of MTBE was added, vortexed for 1 min, placed in an ultrasonic bath for 5 min, and centrifuged at 8700 rpm for 5 min, then the organic phase was removed and the solvent evaporated. The sample was then diluted to 0.2 mL with methanol. 

## 3. Results

### 3.1. Characterization of BM-MSCs and Sensitivity to PTX 

We used three different batches of MSCs isolated from the BM of children who underwent BM collection for sibling transplantation. The characteristics of the donors are reported in Table 1. We verified that the MSC viability remained at > 85% and that the immunophenotype did not change after treatment with 15 µg/mL of PTX for 24 h, as shown in Figure 1. Each BM-MSCs sample was used to produce drug-free secretome (SECR-CTRL) and PTX-loaded secretome (SECR-PTX), respectively, from untreated MSCs and MSCs treated with PTX 15 µg/mL for 24 h.

**Table 1 pharmaceutics-15-02340-t001:** Characteristics of the donors enrolled to produce three batches of the secretome.

ID	SEX	AGE	ID SECRETOME
BM-MSC-1	Male	6 years	SECR-CTRL-1; SECR-PTX-1
BM-MSC-2	Male	9 years	SECR-CTRL-2; SECR-PTX-2
BM-MSC-3	Female	4 years	SECR-CTRL-3; SECR-PTX-3

Subsequently, three OS cell lines, obtained via ATCC (SJSA, MG63, and HOS) were tested with PTX to verify the drug sensitivity. We calculated the PTX IC50 on the three OS cell lines, using an MTT assay after 24 h, and all cell lines were sensitive to PTX. Using the same test, we also calculated the IC_50_ after 5 days of treatment, to compare the effect of free PTX with the PTX content on the EVs. Table 2 summarizes the values for IC_50_, obtained after 24 h and after 5 days of treatment with PTX, in the different OS cell lines, reported in µg/mL and pg/mL, respectively (see Appendix A).

### 3.2. Secretome Cytotoxicity Effect on OS Cell Lines 

After having evaluated that the BM-MSCs were able to withstand high doses of PTX and that the OS cell lines were sensitive to PTX treatment, we analyzed the cytotoxic effect of three batches of SECR-PTX and SECR-CTRL in a 5-day MTT (3-(4,5- dimethyl-2-thiazolyl)-2,5-diphenyl-2-Htetrazoliumbromide) assay on the three OS cell lines, SJSA, MG63, and HOS.

We observed, in all three OS cell lines, that there was a significantly high decrease in viability between untreated cells (control condition—CTRL Viability) and cells treated with 30 µg/mL of PTX (CTRL Mortality), showing a viability of <10% in all OS cell lines. This was also seen with pure SECR-PTX-1-2-3 and with diluted SECR-PTX, showing a dose–response effect. Specifically, we obtained the following mean values with standard deviations in viability: (i) 40.70% ± 16.47 in SJSA, 13.19% ± 2.05 in MG63, 9.55% ± 0.36 in HOS with pure SECR-PTX (*p* < 0.0001 in all cell lines); (ii) 60.15% ± 9.54 in SJSA (*p* < 0.0003), 24.28% ± 6.93 in MG63 (*p* < 0.0001) and 14.49% ± 2.08 in HOS (*p* < 0.0001). With SECR-PTX diluted to 1:2 (*p* < 0.0001), the values were 67.13% ± 8.54 in SJSA (*p* < 0.001), 43.49% ± 13.01 in MG63 (*p* < 0.0001), and 28.62% ± 14.43 in HOS (*p* < 0.0001), respectively. Figure 1 shows all the data obtained for each batch (SECR-1,2,3) in triplicate for the SJSA (panel A), MG63 (panel B ), and HOS (panel C).

### 3.3. Characterization of the Secretome by Nanoparticle Tracking Analysis (NTA)

We investigated the contents of the secretome to verify the presence of EVs through an analysis of the three secretome batches (SECR-1, SECR-2, SECR-3), batches without the drug (SECR-CTRL), and after loading with 15 µg/mL of PTX (SECR-PTX), performed in triplicate by NTA. In Figure 2, we report the NTA results in terms of mean size and size distribution, while in Table 3, we show the Z potential and particle concentration. In scatter mode, the NTA was not able to discriminate the nature of the detected nanoparticles (NPs). In order to discriminate whether the NPs detected in the secretomes were EVs or were a mixture of different biological NPs containing EVs but also other species such as protein aggregates, the samples were stained with a fluorescent membrane marker (CellMask^TM^ Green). Then, the NTA was set to fluorescent mode to exclude the non-stained NPs. Figure 2A,B shows that the size distribution, when measured in scatter mode or in fluorescent mode, did not present significant changes, a good indication that the NPs detected by the NTA were mainly EVs. This does not exclude the presence of other biological species that are not detectable by the NTA, which presents a limit of detection of 30–40 nm. 

Within the two groups, very similar size distributions were detected. The EVs detected in SECR-PTX had a slightly significantly larger size than those in SECR-CTRL (*p*-value < 0.0001). SECR-CTRL presented a mean size of 149 ± 1 nm (DX10: 79.9 nm, DX50: 128.1 nm, DX90: 242.0 nm) while those in SECR-PTX exhibited a mean size of 175 ± 4 nm (DX10: 93.8 nm, DX50: 157.1 nm, DX90: 280.8 nm). In addition, the number of NPs/mL decreased in SECR-PTX compared to those in SECR-CTRL. Finally, the Z potential was negative in all samples. No differences were observed in all the analyzed measures between the different batches. The results that we obtained showed that the cells subjected to treatment with the drug for 24 h produced fewer but larger EVs, compared to the MSCs grown with the standard medium, which produced more but slightly smaller EVs. In response to these data, we assumed that the increased size was due to the presence of PTX, a hypothesis that was evaluated and confirmed by the subsequent HPLC analysis.

### 3.4. High-Performance Liquid Chromatography (HPLC-MS/MS) Analysis

After evaluating the secretome cytotoxicity and the presence of NPs compared to EVs, we verified the presence of PTX in the three SECR batches that were used for the in vitro experiments via HPLC-MS/MS analysis. The presence of PTX was confirmed in all three batches of SECR-PTX with a very low (pg/mL) drug concentration. as showed in Table 4

After the HPLC analysis, we compared the drug contents: (i) using the number of cells obtained after secretome collection (the total volume of the secretome was 20 mL), dividing the content of PTX (pg/mL)/number of cells, we evaluated the PTX content into a single cell (PTX pg/Cell) via proportional calculation; (ii) the number of EVs produced by MSCs was related to collection volume (PTX(pg/mL)/ vol (mL); and (iii) we evaluated the PTX content related to EVs/mL (PTX pg/EVs). The resulting data are reported in Table 5 for each secretome batch. Finally, using the results obtained from the previous analyses, we have demonstrated a linear relationship between PTX concentration in EVs (PTX/EVs Tot ng/mL) and viability (%) in three OS cell lines (SJSA, MG63, HOS) treated with three batches of secretome (SECR-1, SECR-2, and SECR-3), as shown in Figure 3.

## 4. Discussion

In the last years, the association of chemotherapy with radiotherapy and surgery has considerably ameliorated/improved the survival and quality of life of patients affected by OS; however, prognoses remain poor in patients with advanced disease [16].

Moreover, traditional chemotherapy drugs, such as methotrexate (MTX), cisplatin (CIS), doxorubicin (DOX), and ifosfamide (IFO), which are routinely used for OS treatment have some side effects, such as unstable blood concentrations, rapid clearance, poor targeting, and drug resistance. Their use is also limited because they exhibit side effects on normal cells and variable toxicity [17]. For this reason, new strategies must be extended to innovative forms of chemotherapy. To increase the drugs’ therapeutic effect on tumor cells, DDS based on organic and inorganic NPs with a specific target in the tumor have been developed. The MSC secretome, composed of EVs, exosomes, and growth factors, has become an attractive therapeutic agent for some disorders and has been investigated in both preclinical and clinical studies. Recent studies have developed a nano-drug consisting of DOX and MSC-derived exosomes and explored its effect on OS in vitro, resulting in increased cell uptake efficiency and antitumor effect in the OS cell line MG63, as also demonstrated in our experiments [18].

In fact, we have demonstrated that after treatment with PTX, the MSCs achieved the delivery of PTX in association with EVs (microvesicles and exosomes). The secretome thus isolated showed a cytotoxic effect on different tumors, such as mesothelioma, glioblastoma, oral squamous carcinoma, and breast cancer [19].

In this POC study, we demonstrated that the medium that had been conditioned and isolated from the MSCs after PTX treatment contained a new PTX formulation, based on the presence of EVs with a strong cytotoxic effect on OS-tested cell lines.

However, the complexity of the secretome makes it difficult to study and this requires an appropriate approach. The BM-MSC sample variability and the different techniques and media used for cell culturing and isolation can lead to heterogeneity in the samples. Therefore, using a standard method for each phase of the experimental protocol and the decision to select only children as donors allowed us to reduce the sample variability, meaning that all the analyzed tests showed no significant differences between the three batches of secretome (*p* > 0.05). Moreover, we observed that all batches showed cytotoxic activity, regardless of the MSC sample used to produce the secretome. However, we noted a difference in the PTX content according to cell number and the presence of EVs and PTX; therefore, further investigations are needed, and the number of samples that are analyzed needs to be increased. We observed a linear correlation between the content of PTX/EVs Tot and the cytotoxic effect on all cell lines, which exhibited a significant correlation in SJSA and MG63 (*p* < 0.001 and *p* < 0.05, respectively). In the HOS cell line, this correlation was less evident (*p* = 0.07) because the line was more sensitive to PTX. In fact, it was not possible to calculate the PTX-IC_50_ value after 5 days of treatment (the viability in MTT was less than 25% in all conditions, analyzed on a scale from 1000 to 2.5 pg/mL of PTX) as showed in Figure 4.

At the same time, we observed a significant difference (the row factor for cell lines was *p* < 0.0001) between the three OS cell lines, SJSA, MG63, and HOS, after treatment with SECR-PTX-1-2-3. SJSA was less sensitive to PTX (the mean viability was 29.98% ± 14.79 in SJSA, vs 13.00% ± 2.01 in MG63 and 9.67% ± 0.58 in HOS); these data are due to the high heterogeneity of the tumor subtypes. Interestingly, as evidenced by the equal concentrations observed during the standard curve created to obtain the IC_50_ after 5 days, we observed a higher cytotoxic effect of the new PTX formulation in comparison with PTX alone, as shown in Table 2.

It is worth noting that we observed a cytotoxic effect with a very low total concentration of PTX (pg/mL), which was higher than that seen in the samples used for testing free PTX. The cytotoxic effect of the secretome compared to the use of the drug alone has also been confirmed in the literature. These data might support the advantageous use of EVs containing a chemotherapy drug, which acts upon the tumor site through intravenous administration or during surgery, operating directly in the tumor site and reducing the drug’s side effects. In fact, free PTX can cause thrombosis, neuropathy, and high mortality [15,20]; therefore, placing a new formulation of PTX inside the EVs could increase the overall well-being of the patient.

Finally, regarding the physical analysis of the secretome, we observed that the EVs’ mean size (nm) and negative membrane potential were in line with data reported in the literature for the conventional description of EVs [21,22]. The total number of EVs for mL was comparable in three SECR-CTRL batches and in three SECR-PTX batches. These results are encouraging, but future studies will need to isolate the EVs from the conditioned medium obtained from MSC treated with PTX, incorporating a cut-off from a < 100 KDa dialysis filter so as to obtain a secretome to use as an innovative drug delivery system for OS treatment.

## 5. Conclusions

MSC-derived secretome, such as in DDS, can be considered an innovative biotechnological product, constituting an advanced cell-free therapy conveying the drug to the area affected by neoplasia, thereby increasing therapeutic efficacy and limiting the side effects. Since pediatric sarcomas and OS are still very aggressive and difficult to treat, it is interesting to study alternative and new therapies that possibly offer multiple advantages over conventional chemotherapy drugs. In this work, we demonstrated that a conditioned medium obtained from PTX-loaded MSCs contained a new formulation of PTX with a strong cytotoxic effect on OS cell lines. This POC study suggests a new therapeutic approach for OS treatment, with the future objective of producing lyo-secretomeas a stable and easily storable product to use in a standardized GMP-compliant process, along with the creation of a biobank of DDSs ready to use in clinical trials.

## Figures and Tables

**Figure 1 pharmaceutics-15-02340-f001:**
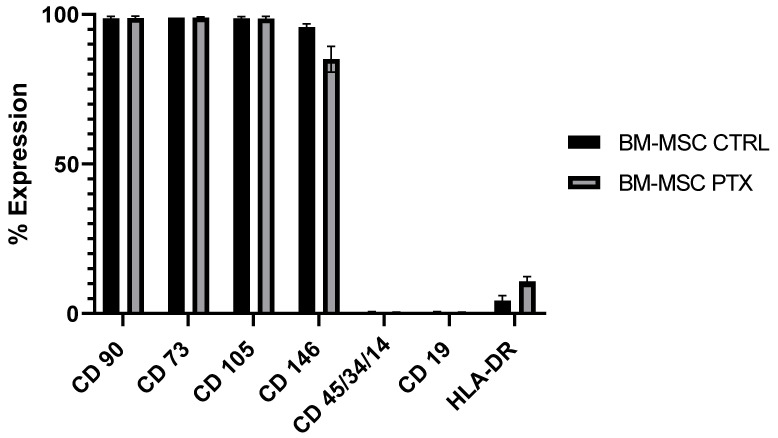
Immunophenotypic analysis of three batches of BM-MSCs, before and after PTX treatment.

**Figure 2 pharmaceutics-15-02340-f002:**
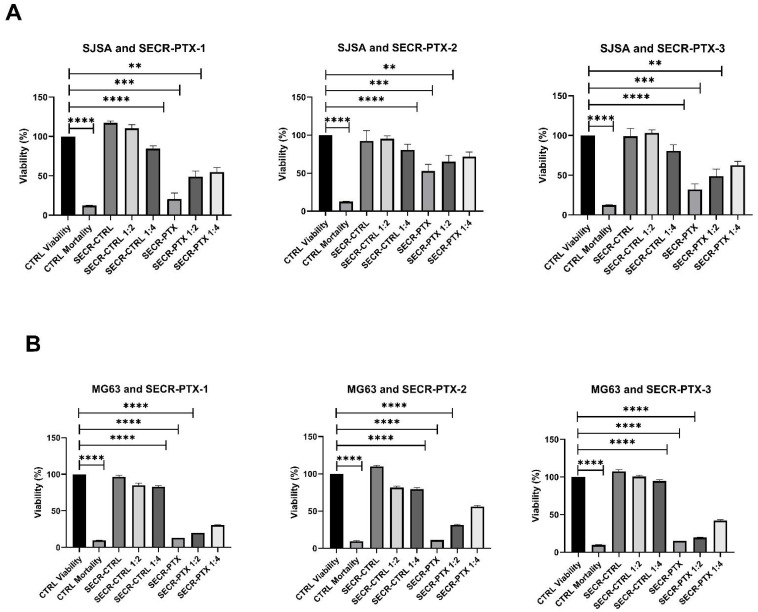
The means of three experiments to perform a viability test with an MTT assay on three OS cell lines: (**A**) SJSA; (**B**) MG63; (**C**) HOS, *n* = 3. CTRL Viability: OS cell lines treated with standard medium; CTRL Mortality: cells treated with 30 µg/mL of PTX; SECR-CTRL: the condition after treatment of three batches of secretome, isolated from untreated MSCs (SECR-CTR-1,2,3), and after the loading of PTX (SECR-PTX-1,2,3), either pure or diluted to 1:2 and 1:4. The symbols **, ***, and **** indicate a significant difference, with *p*-values of <0.01, <0.001, and <0.0001, respectively.

**Figure 3 pharmaceutics-15-02340-f003:**
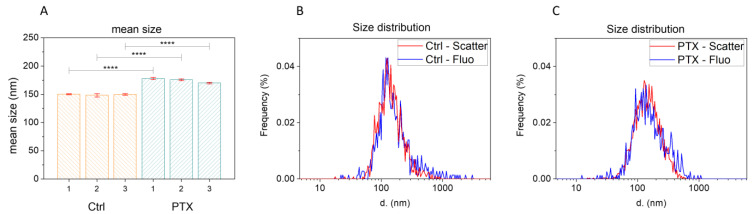
Mean size and size distribution of SECR, obtained via nanoparticle tracking analysis (NTA). (**A**) Mean size distribution of SECR-CTRL-1/2/3 (orange) and SECR-PTX-1/2/3 (green). (**B**) Size distribution of SECR-CTRL-1, not labeled (red) and labeled (blue) with the fluorescent membrane stain, CMG. (**C**) Size distribution of SECR-CTRL-1, both not labeled (red) and labeled (blue) with the fluorescent membrane stain, CMG. The symbols **** indicate a significant difference with a *p*-value of <0.0001.

**Figure 4 pharmaceutics-15-02340-f004:**
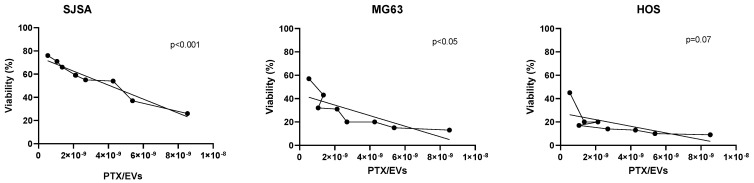
The linear relationship between paclitaxel concentration in EVs (PTX/EVs Tot ng/mL) and viability (%) in three OS cell lines (SJSA, MG63, and HOS), which were treated with three batches of secretome (SECR-1, SECR-2, and SECR-3). The dotted line refers to the quantity of PTX/EVs and the straight line is the trend line.

**Table 2 pharmaceutics-15-02340-t002:** The mean value of the paclitaxel inhibitory concentration (IC_50_) for three OS cell lines (SJSA, MG63, and HOS), reported in µg/mL after 24 h and in pg/mL after 5 days of treatment.

ID Sample	PTX-IC_50_ after 24 h of Treatment	PTX-IC_50_ after 5 Days of Treatment
SJSA	22.8 µg/mL	86.2 pg/mL
MG63	22.6 µg/mL	52.1 pg/mL
HOS	18.6 µg/mL	No detection

**Table 3 pharmaceutics-15-02340-t003:** EVs concentrations (NPs/mL) and Z potentials (mV), measured via nanoparticle tracking analysis.

ID Sample	Particle Concentration (NPs /mL)	Particle Z Potential (mV)
SECR-CTRL-1	2.70 × 10^9^ ± 0.09 × 10^9^	−12.4 ± 3.9
SECR-CTRL-2	2.40 × 10^9^ ± 0.05 × 10^9^	−36.4 ± 2.6
SECR-CTRL-3	2.50 × 10^9^ ± 0.09 × 10^9^	−15.5 ± 3.7
SECR-PTX-1	0.51 × 10^9^ ± 0.02 × 10^9^	−13.6 ± 2.8
SECR-PTX-2	0.74 × 10^9^ ± 0.02 × 10^9^	−17.5 ± 8.7
SECR-PTX-3	0.49 × 10^9^ ± 0.01 × 10^9^	−26.3 ± 5.3

**Table 4 pharmaceutics-15-02340-t004:** High-performance liquid chromatography analysis of three batches of secretome (SECR-CTRL without the drug, SECR-PTX with the drug). The PTX concentration is reported in pg/mL.

ID SECRETOME	PTX Concentration pg/mL
SECR-CTRL-1	no detection
SECR-CTRL-2	no detection
SECR-CTRL-3	no detection
SECR-PTX-1	86.05
SECR-PTX-2	31.03
SECR-PTX-3	52.38

**Table 5 pharmaceutics-15-02340-t005:** The summary data for three batches of secretome (SECR-PTX-1; SECR-PTX-2; SECR-PTX-3).

ID Sample	No. of Cells (Tot)	% MSC Viability	No. of Cells/SECR-mL	No. of EVs (Tot)	No. of EVs/mL	No. of EVs/Cells	Content of PTX (pg/mL)	Content of PTX/Cell (pg/cell)	Content of PTX/EVs (pg/EVs)
SECR-PTX-1	1.92 × 10^6^	85.7%	9.60 × 10^4^	1.01 × 10^10^	5.07 × 10^8^	5277.8	86.5	9.01 × 10^−4^	1.71 × 10^−7^
SECR-PTX-2	9.07 × 10^5^	94.6%	4.53 × 10^4^	1.47 × 10^10^	7.37 × 10^8^	16,250.0	31.0	6.84 × 10^−4^	4.21 × 10^−8^
SECR-PTX-3	1.86 × 10^6^	94.9%	9.30 × 10^4^	9.73 × 10^9^	4.87 × 10^8^	5233.0	52.4	5.63 × 10^−4^	1.08 × 10^−7^

## Data Availability

All data generated or analyzed during this study are included in this published article. However, the data obtained in this study are also available from the corresponding author upon request.

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
