# Peer review of "A New Paclitaxel Formulation Based on Secretome Isolated from Mesenchymal Stem Cells Shows a Significant Cytotoxic Effect on Osteosarcoma Cell Lines"

_pharmaceutics, 2023, doi:10.3390/pharmaceutics15092340_

Round 1
Reviewer 1 Report
The current manuscript investigates the cytotoxic effect of the secretome derived from MSCs treated with paclitaxel on 3 different osteosarcoma cell lines. Data points to a strong cytotoxic effect of 3 different secretome batches at very low concentrations of paclitaxel. The data also suggests the presence of EVs in the secretome. The following aspects need to be addressed before publication:
Major:
1. Although the data suggests the presence of EVs, further controls are required to prove the presence of vesicles according to the guidelines of the ISEV (consider western blot for EV markers, TEM).
2. The presence of EVs in the secretome does not immediately prove that these EVs are responsible for the cytotoxic effect. This confirmation could be achieved by using EV-depleted secretomes.
3. Figure 1. Error bars need to be added to the viability data. How many times were the experiments repeated with each batch. In addition, the meaning of “CTRL Mortality“ needs to be indicated in the figure legend.
4. Wei et al., (PMID 35959282) investigated the delivery of doxorubicin to osteosarcoma cells by MSC-EVs. The authors should address these findings in the discussion and compare them with their own findings.
Minor:
5. Table 2 (line 124). It is surprising that the IC50 increases with longer treatment times. Reasons for this outcome should be proposed in the discussion section.
6. Line 329. There is a typo in the composition of the eluent A. Please, check.
7. The table in lines 182-183 has to be relabeled as table 3. Furthermore, the headings “EV concentration“ and “EV Z potential“ should be replaced by “Particle concentration“ and “Particle Z potential“.
Author Response
Dear reviewer ,
please find enclosed the revised manuscript entitled “A New Paclitaxel Formulation Based On Secretome Isolated from Mesenchymal Stem Cells Shows a Significant Cytotoxic Effect on Osteosarcoma Cell Lines “
We wish to thank you for giving us the opportunity to revise and improve our work.
Moreover, the whole manuscript has been reviewed by a native English speaker.
In this revised version we took into consideration your reviewers’ criticisms and comments and modified the text accordingly.
All alterations in the revised manuscript are tracked using the "Track Changes" function in Microsoft Word.
We added a figure (the figure 1 ) in the manuscript and one in the supplementary materials and modified the number of the figures in the text.
We believe that following the reviewers’ suggestions the new version has been much improved, and hope that it is now suitable for publication in your journal.
Please, contact us with no hesitation for any further questions or needed modifications.
Yours sincerely
Katia Mareschi
Below is our response point to point
The current manuscript investigates the cytotoxic effect of the secretome derived from MSCs treated with paclitaxel on 3 different osteosarcoma cell lines. Data points to a strong cytotoxic effect of 3 different secretome batches at very low concentrations of paclitaxel. The data also suggests the presence of EVs in the secretome. The following aspects need to be addressed before publication:
Major:
- Although the data suggests the presence of EVs, further controls are required to prove the presence of vesicles according to the guidelines of the ISEV (consider western blot for EV markers, TEM).
In this paper our aim was to verify if the conditioned medium from MSCs loaded with Paclitaxel (PTX) contained a secretome with an antitumor effect on three OS cell lines (SJSA, MG-63 and HOS) as a proof of concept. We demonstrated that it had a strong cytotoxic effect and showed the presence of microvesicles through NTA analysis and HPLC analysis. We know that the presence of EVs should be investigated and analyzed using other methodologies, which are now ongoing to improve our study so to submit a new manuscript with new data describing the isolation of EVs, their characteristics, and their cytotoxic effects. The guidelines of the ISEV are the starting point for this new study.
- The presence of EVs in the secretome does not immediately prove that these EVs are responsible for the cytotoxic effect. This confirmation could be achieved by using EV-depleted secretomes.
This experiment is also scheduled for the next research study design.
- Figure 1. Error bars need to be added to the viability data. How many times were the experiments repeated with each batch. In addition, the meaning of “CTRL Mortality“ needs to be indicated in the figure legend.
As suggested, we added the error bars in the revised version of the manuscript
- Wei et al., (PMID 35959282) investigated the delivery of doxorubicin to osteosarcoma cells by MSC-EVs. The authors should address these findings in the discussion and compare them with their own findings.
We reviewed the discussion considering this work.
Minor:
- Table 2 (line 124). It is surprising that the IC50 increases with longer treatment times. Reasons for this outcome should be proposed in the discussion section.
In table 2 the IC50 at 24 h is reported in microg/ml and after 5 days is reported in pg/ml as indicated in the column header. For greater clarity, in the reviewed manuscript we indicated near the values also the units.
- Line 329. There is a typo in the composition of the eluent A. Please, check.
We modified it.
- The table in lines 182-183 has to be relabeled as table 3. Furthermore, the headings “EV concentration“ and “EV Z potential“ should be replaced by “Particle concentration“ and “Particle Z potential“.
We modified it.

Reviewer 2 Report
The manuscript presents a study focused on exploring the potential of a secretome derived from Mesenchymal Stem Cells treated with Paclitaxel for targeted drug delivery in Osteosarcoma treatment.
Comments:
1. Maintaining MSC Phenotype after PTX: The authors should clarify how they ensured that PTX treatment did not alter the fundamental characteristics (phenotype) of the MSCs. This is important for maintaining the intended therapeutic properties of the secretome.
2. In the cell viability assay, authors should include PTX control (1:2, 1:4) for the comparison with SECR-PTX.
3. In the figure 1 error bar is missing also author should mention the number of replicates in the figure legend.
4. Author should provide the IC50 value data graph in the supplementary for the all the cell lines.
5. In the table 3: SECR-PTX-1 PTX concentration pg/ml should be verified (86,05?)
6. Table 4: what is the + sign indicate?
7. Figure 3 need to be quote in the results section. Author should mention about how many replicates were used in the finding linear correlation and missing.
8. In the line 294: no need to mentions the results in the method sections.
Need to be improved.
Author Response
Dear reviewer ,
please find enclosed the revised manuscript entitled “A New Paclitaxel Formulation Based On Secretome Isolated from Mesenchymal Stem Cells Shows a Significant Cytotoxic Effect on Osteosarcoma Cell Lines “
We wish to thank you for giving us the opportunity to revise and improve our work.
Moreover, the whole manuscript has been reviewed by a native English speaker.
In this revised version we took into consideration your reviewers’ criticisms and comments and modified the text accordingly.
All alterations in the revised manuscript are tracked using the "Track Changes" function in Microsoft Word.
We added a figure (the figure 1) in the manuscript and one in the supplementary materials and modified the number of the figures in the text.
We believe that following the reviewers’ suggestions the new version has been much improved, and hope that it is now suitable for publication in your journal.
Please, contact us with no hesitation for any further questions or needed modifications.
Yours sincerely
Katia Mareschi
Below is our response point to point:
The manuscript presents a study focused on exploring the potential of a secretome derived from Mesenchymal Stem Cells treated with Paclitaxel for targeted drug delivery in Osteosarcoma treatment.
Comments:
- Maintaining MSC Phenotype after PTX: The authors should clarify how they ensured that PTX treatment did not alter the fundamental characteristics (phenotype) of the MSCs. This is important for maintaining the intended therapeutic properties of the secretome.
In a previous manuscript (Lisini et al, 2020) we demonstrated that MSCs isolated from adipose tissue after PTX treatment didn't change immunophenotype. Thanks to the reviewer's comments we performed additional experiments and we have added figure 1 which shows the immunophenotype before and after PTX treatment obtained on the 3 batches of BM-MSCs used for this manuscript. We modified the number of the figures in the text.
- In the cell viability assay, authors should include PTX control (1:2, 1:4) for the comparison with SECR-PTX.
In our experimental conditions we used a PTX dose which induced a viability <10% after 5 days of treatment. In our preliminary experiments we also used the supernatant of MSCs after loading with 15 µg/ml before secretome production to verify if the PTX in the supernatant not up-taken from MSCs had a cytotoxic effect comparable to free PTX. We also diluted this supernatant in 1:2 and 1:4 and no significant differences were observed in comparison with the free PTX. Moreover, HPLC analyses showed that only a small quantity of PTX was uptaken by MSCs and the secretome contained this very small quantity of PTX. We did not include these preliminary results in the manuscript because we focused our experiments on the secretome which showed a comparable cytotoxic effect to high dose of PTX after 5 days of treatment.
In the word file you can see the preliminary results.
The conditions 1, 2, 3 and 4 are respectively:
1) pure PTX (30 µg/ml);
2) supernatant of the MSC medium after treatment with PTX 15 µg/ml before secretome production
3) condition 2 diluted 1:2
4) condition 2 diluted 1:4
No significant differences were observed between the 4 conditions (2 ways-ANOVA test show a p > 0.05 in multi comparison analysis).
- In the figure 1 error bar is missing also author should mention the number of replicates in the figure
In the material and method section we have explained that all experiments were performed in triplicate and we added the error bars in the graph.
- Author should provide the IC50 value data graph in the supplementary for the all the cell lines.
In the Supplementary Appendix we have added the graphs obtained on GraphPad to evaluate the IC50 for each cell line.
- In the table 3: SECR-PTX-1 PTX concentration pg/ml should be verified (86,05?)
We have modified it.
- Table 4: what is the + sign indicate?
We have eliminated the symbols that could create confusion.
- Figure 3 need to be quote in the results section. Author should mention about how many replicates were used in the finding linear correlation and missing.
We have quoted this figure (now figure 4) in the discussion session.
- In the line 294: no need to mentions the results in the method sections.
We have modified the text as suggested from the reviewer.
Comments on the Quality of English Language
Need to be improved.
The whole manuscript has been reviewed by a native English teacher.

Reviewer 3 Report
In their manuscript entitled "A New Paclitaxel Formulation Based On Secretome Isolated from Mesenchymal Stem Cells Shows a Significant Cytotoxic Effect on Osteosarcoma Cell Lines" the authors describe an very interisting research on an umet need in cancer therapy.
In general, this ia a very interesting manuscript and the research design was appriptiately choosen.
I have the following suggestions to improve the quality of the manuscript:
- Abstract: The Abstract should be written in passive style and needs aome grammatical workup (e.g. line 32 : "The HPLC analyses detected the presence of PTX in minimal doses in all SECR batches"). So I suggest to check the abtract on it soundness and rephrase it (so it will become an "eye catcher" and is easier to read)
- Introduction. Very interesting would be to add also some more side effects of cytostatica; what are promiment and less prominent side effects; for example can Paclitaxel also cause thrombosis - which would be a significant side effect of high mortality - see for example latest reference on side effect for cyclophosphamide : Cells 2023, 12(15), 1965; https://doi.org/10.3390/cells12151965
- Figures: Add error bars, add for better understanding the p-values directly behind the *
- Methods: line 311: two-way anova est (Figure 1). witrie anova in Capitals: ANOVA
In general the Methods part need some stylish workup for more fluent reading.
To summarize it is a very interesting manuscript.
needs workup by native English speaker
Author Response
Dear reviewer ,
please find enclosed the revised manuscript entitled “A New Paclitaxel Formulation Based On Secretome Isolated from Mesenchymal Stem Cells Shows a Significant Cytotoxic Effect on Osteosarcoma Cell Lines “
We wish to thank you for giving us the opportunity to revise and improve our work.
Moreover, the whole manuscript has been reviewed by a native English speaker.
In this revised version we took into consideration your reviewers’ criticisms and comments and modified the text accordingly.
All alterations in the revised manuscript are tracked using the "Track Changes" function in Microsoft Word.
We added a figure (the figure 1) in the manuscript and one in the supplementary materials and modified the number of the figures in the text.
We believe that following the reviewers’ suggestions the new version has been much improved, and hope that it is now suitable for publication in your journal.
Please, contact us with no hesitation for any further questions or needed modifications.
Yours sincerely
Katia Mareschi
Below is our response point to point
In their manuscript entitled "A New Paclitaxel Formulation Based On Secretome Isolated from Mesenchymal Stem Cells Shows a Significant Cytotoxic Effect on Osteosarcoma Cell Lines" the authors describe an very interisting research on an umet need in cancer therapy.
In general, this ia a very interesting manuscript and the research design was appriptiately choosen.
Thank you to the reviewer for the positive comments.
I have the following suggestions to improve the quality of the manuscript:
- Abstract: The Abstract should be written in passive style and needs aome grammatical workup (e.g. line 32 : "The HPLC analyses detected the presence of PTX in minimal doses in all SECR batches"). So I suggest to check the abtract on it soundness and rephrase it (so it will become an "eye catcher" and is easier to read)
We have modified it.
- Introduction. Very interesting would be to add also some more side effects of cytostatica; what are promiment and less prominent side effects; for example can Paclitaxel also cause thrombosis - which would be a significant side effect of high mortality - see for example latest reference on side effect for cyclophosphamide : Cells 2023, 12(15), 1965; https://doi.org/10.3390/cells12151965
We argued this point in the discussion and in the reviewed part we also added the reference suggested by the reviewer.
- Figures: Add error bars, add for better understanding the p-values directly behind the *
We have added them in the reviewed version.
- Methods: line 311: two-way anova est (Figure 1). witrie anova in Capitals: ANOVA
We have modified the text.
In general the Methods part need some stylish workup for more fluent reading.
The whole manuscript has been reviewed by a native English teacher.
To summarize it is a very interesting manuscript.
Thank you and we hope that the reviewed manuscript is now acceptable for publication in Pharmaceutics.

Round 2
Reviewer 1 Report
The authors have addressed all my concerns and delivered an improved version.
Reviewer 2 Report
NA
NA